# Common Neighbor Induced Message Passing for Inductive Link Prediction in Knowledge Graphs

## Abstract

Inductive link prediction is a significant challenge in knowledge graphs, focusing on predicting potential relations between unseen entities during training. A promising approach is to utilize Graph Neural Networks (GNNs) to extract entity-independent features from surrounding subgraphs. However, existing mainstream subgraph extraction methods may lead to the loss of key entities and relations, resulting in many disconnected reasoning paths that seriously hinder effective message passing. To address this challenge, we propose a novel framework called Common Neighbor Induced Message Passing (CNMP), designed to enhance message passing even when reasoning paths are disconnected. We observe that the common neighbors of two entities must share a reasoning path. Based on this insight, CNMP enhances message passing by updating the distance labels of isolated common neighbors, even if they are unreachable. This allows CNMP to incorporate new connected equivalent relations, facilitating effective message passing. Furthermore, we introduce a $CNMP_+$ strategy that further improves the preservation of entities and relations during the message-passing process. $CNMP_+$ involves maintaining a list of common neighbors at various distances and using a probing strategy to reconstruct complete reasoning paths. Experiments across multiple datasets demonstrate that our method significantly outperforms existing state-of-the-art methods.

## 1 Introduction

Knowledge graphs organize human knowledge in the form of factual triples (head entity, relation, tail entity), and these graphs represent entities as nodes and relations as edges. Examples of knowledge graphs include WordNet (Miller, 1995), Freebase (Bollacker et al., 2008), and DBPedia (Auer et al., 2007). Recently, knowledge graphs have been widely used in natural language processing (Zhang et al., 2019), question answering (Huang et al., 2019), and recommendation systems (Wang et al., 2018). However, real-world knowledge graphs confront the challenge of continuously emerging new entities, such as new users and products in e-commerce knowledge graphs or new molecules in biomedical knowledge graphs(Breit et al., 2020; Bonner et al., 2022). Moreover, knowledge graphs often suffer from incompleteness, i.e., some links are missing.

To address these challenges, extensive research efforts have been devoted to the inductive link prediction task (Tran et al., 2016; Hamaguchi et al., 2017; Chamberlain et al., 2022; Lin et al., 2022). Inductive link prediction intends to predict missing links between entities in knowledge graphs, where entities during the training and inference stages can be different. Despite the importance of inductive link prediction in real-world applications, many existing knowledge graph completion methods focus on the transductive link prediction task (Bordes et al., 2013; Lin et al., 2015b; Trouillon et al., 2016; Yang et al., 2015), which can only handle the entities seen during the training stage(Sadeghian et al., 2019). Inductive link prediction is challenging because it requires models to predict missing relations between unseen entities during training. This means that models need to generalize what they have learned from training entities to unseen entities.

Existing inductive link prediction methods mainly focus on extracting surrounding subgraphs around the relations to be predicted and learning entity-independent features within these subgraphs, such as relational representations and relational correlations. Graph Neural Network (GNN)-based approaches

are well-known for their effective message-passing mechanism (e.g., graph convolution) that leverages the structure of graphs to extract features within subgraphs. For example, GraIL (Teru et al., 2020) is the first to introduce a GNN-based framework for relation prediction, which models relational representations and reasons over subgraphs. CoMPILE (Mai et al., 2021) updates both relation and entity embeddings to improve entity-relation interaction during message-passing. TACT (Chen et al., 2021) focuses on modeling semantic correlations of relations within subgraphs, which uses topology-aware correlations to boost edge-level interactions.

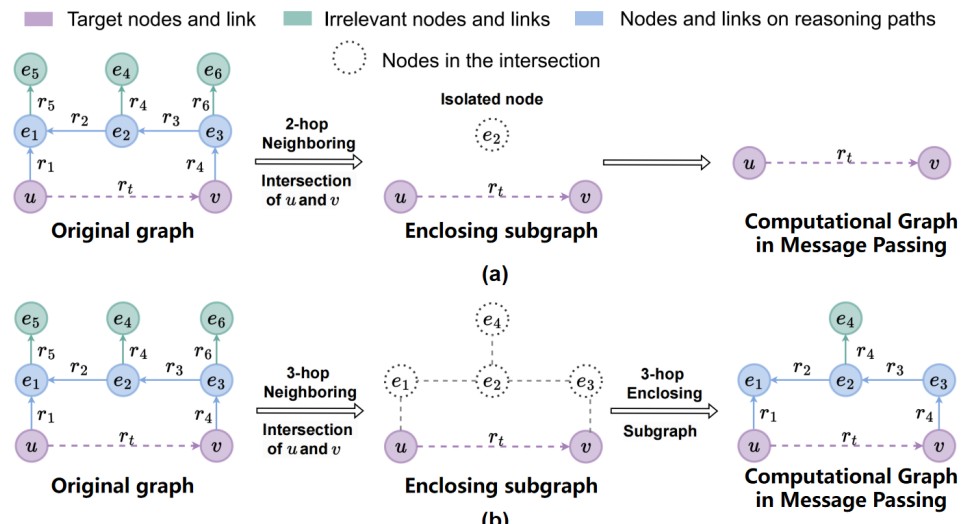

Figure 1: An example of the message passing process on the extracted enclosing subgraph. Enclosing subgraph contains links between nodes in the $n$-hop neighborhood intersection of target nodes.

The above methods are mainly based on enclosing subgraphs (Teru et al., 2020; Mai et al., 2021; Geng et al., 2022; Lin et al., 2022; Xu et al., 2022; Chen et al., 2021), which contain relations in the $n$-hop neighborhood intersection of the target entities(the head and tail of the relation to be predicted). However, key entities and relations may be lost during the process of extracting enclosing subgraphs, causing a significant amount of disconnected reasoning paths. This severely hinders message passing for GNN-based methods, and consequently impacts overall reasoning performance. As shown in Figure 1, the extraction of the 2-hop enclosing subgraph induced by entities $u$ and $v$ results in the drop of key entities $e_1$ and $e_3$, leading to the loss of relations $r_2$ and $r_3$, leading to the disconnected reasoning paths of the isolated node $e_2$. The 3-hop enclosing subgraph can preserve the reasoning path, while it introduces irrelevant relations $r_4$ and nodes $e_4$ as well. On the WN18RR dataset (Toutanova & Chen, 2015), over 45% of 2-hop enclosing subgraphs extract only the target relations, and over 75% of 3-hop enclosing subgraphs contain irrelevant relations (see Section 2.2).

To tackle this problem, we observe that the common neighbors of two entities must share a reasoning path (please refer to Appendix A for detailed proof). Based on this insight, we propose a novel Common Neighbor Induced Message Passing (CNMP) framework. Specifically, without introducing any irrelevant information, we propose CNMP strategy to restore the connectivity of the disconnected reasoning paths, and further propose CNMP$_+$ strategy to iteratively reconstruct the disconnected reasoning paths.

(i) *CNMP strategy*. Given that the message passing mechanism relies on the distance information between nodes to select the next passing node. To ensure that isolated common neighbors on disconnected reasoning can perform message passing, *CNMP strategy* update their distance to the target node from infinity to their shortest distance on the original graph. Moreover, to preserve as much information from the original reasoning path as possible, we introduce a new equivalence relation (the original relation with added distance information) as the content of the message passing for the isolated common neighbor.

(ii) *CNMP$_+$ strategy*. First, *CNMP$_+$ strategy* maintains a list of common neighbors at varying distances from the target node as the data foundation for probing. Then, through a target-driven probing strategy, *CNMP$_+$* starts with isolated common neighbors and systematically

explores lists of common neighbors closer to the target node. During this process, only the current most probable nodes for the reasoning path are retained while the rest are discarded, thereby iteratively reconstructing the original reasoning path with precision.

Inspired by (Chen et al., 2021), we integrate two classic GNN-based methods, RCN and RGCN, with the novel common neighbor induced message passing strategies, named respectively as CNMP-base and CNMP models. Experiments demonstrate that CNMP method accurately extracts key information from subgraphs based on a new messaging mechanism, leading to a superior performance over existing state-of-the-art methods on inductive link prediction. Moreover, experiments under large KGs and fully inductive settings demonstrate a superior scalability and generalization ability of CNMP method.

## 2 METHODS

### 2.1 MOTIVATION

To further elaborate on our motivation, we introduce a rule-based learning perspective to analyze the reasoning process within the subgraph. Rule-based approaches seek to extract first-order logical rules by analyzing reasoning paths within knowledge graphs. Each rule comprises a head and a body, with the head representing a single atom, i.e., a fact expressed in the form of *Relation(head entity, tail entity)*, while the body is a set of atoms. Given a head $R(y, x)$ and body $B_1(y, z_1) \wedge B_2(z_1, z_2) \wedge \cdots \wedge B_T(z_{T-1}, x)$, there is a rule $R(y, x) \leftarrow B_1(y, z_1) \wedge B_2(z_1, z_2) \wedge \cdots \wedge B_T(z_{T-1}, x)$. A rule is connected when each atom within it shares a common variable with at least one other atom, and a rule is closed if each variable in the rule appears in at least two atoms (Yang et al., 2017).

In particular, we pay more attention to the closed and connected rules, that is, the reasoning paths between target nodes, as closeness and connectedness prevent finding rules with irrelevant relations (Sadeghian et al., 2019). **These closed and connected paths are also crucial for ensuring the effectiveness of message passing and feature extraction**. We define the length of a rule as the count of nodes forming the rule, and **define the relevant rules** as those that are both closed, connected, and with a length not exceeding $2h + 1$. The irrelevant rules are those that are either not closed, not connected, or possess a length exceeding $2h + 1$, where $h$ denotes the number of hops for the neighboring nodes of the target nodes. To align with prior work (Teru et al., 2020), we set the hop value as 2.

Existing methods mainly reason on enclosing subgraphs (Teru et al., 2020; Mai et al., 2021; Geng et al., 2022; Lin et al., 2022; Xu et al., 2022; Chen et al., 2021). The original intention of the enclosing subgraph is twofold: to effectively *eliminate the irrelevant rules* and *preserve the relevant rules* around the target link. However, the enclosing subgraph cannot achieve both criteria. Specifically, the 2-hop enclosing subgraph produces disconnected reasoning paths, hindering message passing and failing to preserve relevant rules. Conversely, the 3-hop enclosing subgraph allows effective message passing but introduces irrelevant rules, leading to lower-quality features extracted through message passing.

We propose two toy examples as illustrations and further conduct statistical analyses to reveal the prevalence of the phenomenon in Section 2.2.

### 2.2 ANALYSIS OF THE ENCLOSING SUBGRAPH

The basic steps for subgraph extraction are as follows: For a triple $(u, r_t, v)$, the initial step involves the extraction of the enclosing subgraph around the target nodes $u$ and $v$ (Teru et al., 2020). The subsequent steps outline the procedure for generating the enclosing subgraph between nodes $u$ and $v$. First, we compute the neighborhoods $\mathcal{N}_k(u)$ and $\mathcal{N}_k(v)$ for $u$ and $v$, respectively. Here, $k$ denotes the maximum distance used to define the neighborhoods around nodes $u$ and $v$. Second, we take an intersection of $\mathcal{N}_k(u)$ and $\mathcal{N}_k(v)$ to get $\mathcal{N}_k(u) \cap \mathcal{N}_k(v)$. Third, we calculate the enclosing subgraph $\mathcal{G}(u, r_t, v)$ by removing isolated nodes from the intersection of sets $\mathcal{N}_k(u)$ and $\mathcal{N}_k(v)$, where isolated nodes are defined as nodes lacking any edges connecting them to other nodes within $\mathcal{N}_k(u) \cap \mathcal{N}_k(v)$.

Now we analyze the issues with the subgraph.

Table 1: Statistics on inductive datasets when setting the neighbor hop $h$ to 2. The values on the right of Num = 2, Num = 3, and Others denote the proportion of the corresponding type of subgraphs to the total number of extracted 2-hop enclosing subgraphs of each dataset. Num = 2 denotes that the extracted enclosing subgraph only consists of the head entity $u$ and the tail entity $v$ with the target relation between them. Num = 3 denotes that apart from the head entity $u$ and the tail entity $v$, the extracted enclosing subgraph only remains one other entity consisting of the path from $u$ to $v$. The Incomplete_Ratio is the proportion of the total nodes in the enclosing subgraph to the number of nodes in the enclosing subgraph with CNMP$_+$ strategy. The smaller the Incomplete_Ratio value is, the more relevant rule loss is caused by the 2-hop enclosing subgraph extraction method.

| | | WN18RR | | | | FB15k-237 | | | | NELL-995 | | | |
| --- | --- | --- | --- | --- | --- | --- | --- | --- | --- | --- | --- | --- | --- |
| | | v1 | v2 | v3 | v4 | v1 | v2 | v3 | v4 | v1 | v2 | v3 | v4 |
| Statistics in Training Set | Num=2 | 0.505 | 0.453 | 0.463 | 0.471 | 0.134 | 0.125 | 0.092 | 0.056 | 0.318 | 0.168 | 0.074 | 0.105 |
| | Num=3 | 0.073 | 0.082 | 0.086 | 0.087 | 0.024 | 0.018 | 0.014 | 0.003 | 0.001 | 0.001 | 0.001 | 0.002 |
| | Others | 0.422 | 0.465 | 0.450 | 0.465 | 0.847 | 0.859 | 0.859 | 0.941 | 0.682 | 0.832 | 0.894 | 0.894 |
| | Incomplete_Ratio | 0.349 | 0.351 | 0.342 | 0.353 | 0.531 | 0.634 | 0.698 | 0.754 | 0.362 | 0.566 | 0.639 | 0.648 |
| Statistics in Testing Set | Num=2 | 0.528 | 0.505 | 0.504 | 0.510 | 0.090 | 0.049 | 0.092 | 0.062 | 0.525 | 0.221 | 0.174 | 0.174 |
| | Num=3 | 0.034 | 0.029 | 0.039 | 0.042 | 0.004 | 0.002 | 0.014 | 0.004 | 0.014 | 0.003 | 0.006 | 0.001 |
| | Others | 0.456 | 0.481 | 0.477 | 0.443 | 0.906 | 0.941 | 0.897 | 0.930 | 0.469 | 0.776 | 0.822 | 0.827 |
| | Incomplete_Ratio | 0.564 | 0.579 | 0.594 | 0.576 | 0.811 | 0.862 | 0.897 | 0.873 | 0.685 | 0.763 | 0.843 | 0.837 |

From a rule mining perspective, reasoning on the subgraph based on the massage pass strategy can be regarded as a similar process to extracting rules from the subgraph.

The 2-hop enclosing subgraph method suffers from effectively preserving relevant rules. As depicted in Figure 1a, when extracting the 2-hop enclosing subgraph from the original knowledge graph, it eliminates all other entities and their corresponding relations. To further elaborate on the prevalence of the observation, we conduct a statistical analysis by setting the neighborhood hop as 2 across the inductive benchmarks in Table 1. From Table 1, we can see that the enclosing subgraph extraction method incurs a significant loss of the relevant rules. Specifically, for WN18RR(Toutanova & Chen, 2015), nearly half of the extracted subgraphs only contain the target nodes $u$ and $v$. The Incomplete_Ratio also supports the statement. Varied degrees of relevant rule loss are also discernible in FB15K-237(Dettmers et al., 2018) and NELL-995(Xiong et al., 2017). This may cause a significant loss of relevant nodes and relations, which may severely hinder message passing.

Table 2: Statistics on inductive datasets when setting the neighbor hop $h$ to 3. The values below each version of inductive datasets denote the proportion of the extracted 3-hop enclosing subgraphs that contain irrelevant rules to the total number of extracted 3-hop enclosing subgraphs.

| | WN18RR | | | | FB15k-237 | | | | NELL-995 | | | |
| --- | --- | --- | --- | --- | --- | --- | --- | --- | --- | --- | --- | --- |
| | v1 | v2 | v3 | v4 | v1 | v2 | v3 | v4 | v1 | v2 | v3 | v4 |
| Statistics in Training Set | 0.872 | 0.756 | 0.759 | 0.850 | 0.979 | 0.988 | 0.989 | 0.992 | 0.969 | 0.998 | 0.999 | 0.998 |
| Statistics in Testing Set | 0.580 | 0.501 | 0.575 | 0.544 | 0.615 | 0.787 | 0.798 | 0.830 | 0.989 | 0.918 | 0.883 | 0.917 |

The 3-hop enclosing subgraph suffers from eliminating the irrelevant rules. As shown in Figure 1b, when extracting the 3-hop enclosing subgraph from the original knowledge graph, the 3-hop enclosing subgraph cannot eliminate the irrelevant nodes $e_7$, $e_8$, $e_9$, and their corresponding relations. From Table 2, we can see that the majority of the 3-hop enclosing subgraphs contain irrelevant rules. This may let the model overfit to the irrelevant nodes and relations, which affect the quality of features extracted through message passing.

Therefore, to tackle these problems, we propose a novel Common Neighbor Induced Message Passing framework (CNMP), to ensure effective message passing even when reasoning paths in enclosing subgraphs are disconnected, and to avoid introducing irrelevant information. Based on the observation that the common neighbors of two entities must be on a certain reasoning path (please refer to Appendix A for details), our framework introduces two strategies starting from common neighbors. Specifically, for disconnected reasoning paths, we propose CNMP strategy to efficiently restore path connectivity, and further introduce CNMP$_+$ strategy to iteratively reconstruct the accurate and complete paths.

Inspired by (Chen et al., 2021), we integrate two classic GNN-based methods with our CNMP strategies, RCN and RGCN, both of which learn features for prediction based on enclosing subgraphs

employing a GNN-like message-passing mechanism that aggregates neighbor information. To further clarify the relationship between CNMP, RCN, and RGCN, Figure 2 illustrates how the CNMP strategy tackles the challenge of dealing with enclosing subgraphs that contain numerous disconnected paths. During the message-passing process employed by RCN and RGCN, it introduces equivalent relations or reconstructs reasoning paths to ensure the smooth progression of information exchange. This guarantees the effectiveness of message passing despite the presence of gaps in the subgraphs' connectivity.

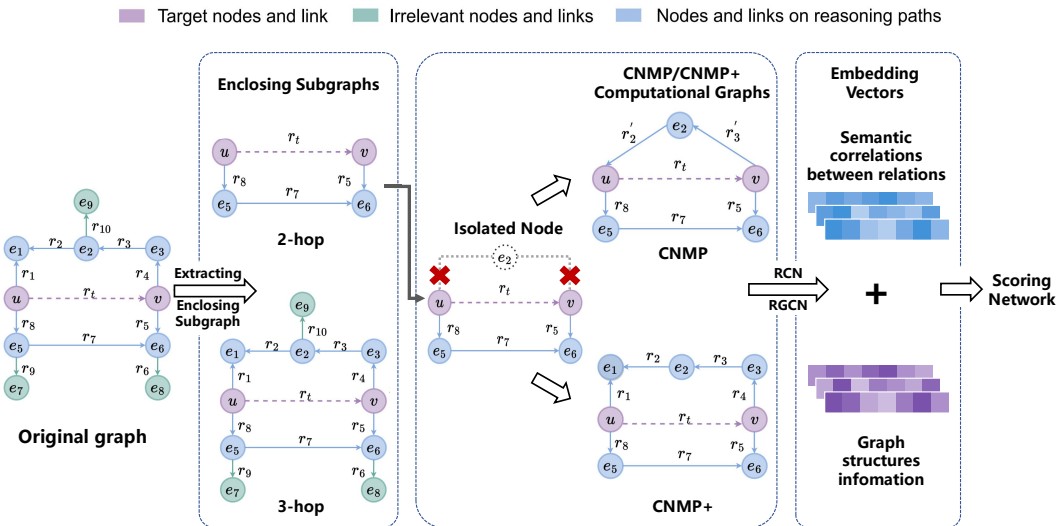

Figure 2: A comparative analysis between 2-hop enclosing subgraph with CNMP strategy and 3-hop enclosing subgraph with CNMP$_+$ strategy during the message-passing process employed by RCN and RGCN.

## 2.3 THE CNMP STRATEGY

The enclosing subgraph method may result in the problem of disconnected reasoning paths, as in Figure 2 the reasoning paths between node $e_2$ and target nodes $u$ and $v$ are disconnected. Therefore, we propose CNMP strategy to restore disconnected reasoning paths in enclosing subgraphs.

Specifically, for isolated nodes, we use the shortest path distances in the original knowledge graph to update their unreachable distance labels, so the isolated nodes can still pass messages. Moreover, to explicitly distinguish the actual connection situation between the nodes in the common neighborhood and the target nodes $u$ and $v$, we directly incorporate the distance information into the relation representation of the node for message passing. Specifically, this is reflected in the vector concatenation operation. For example, in the 2-hop enclosing subgraph of Figure 2, for the node $e_2$ that is an isolated common neighborhood, the CNMP strategy replaces the relations $r_2$ and $r_3$ with new equivalent relations to the target nodes. Specifically, $r_2^{'} = r_2 \oplus (d(e_2, u), d(e_2, v))$ and $r_3^{'} = r_3 \oplus (d(e_2, u), d(e_2, v))$, where $\oplus$ denotes vector concatenation, and $d(i, u)$ denotes the shortest distance between nodes $i$ and $u$ in the original knowledge graph without considering any path through node $v$.

As shown in Figure 2, based on CNMP strategy, we effectively restore disconnected reasoning paths between node $e_2$ and targets nodes $u$ and $v$, and we also effectively eliminate nodes and corresponding relations that are not on any reasoning path, thereby improving the accuracy and efficiency of message passing. For a clearer and more detailed process, please refer to the pseudo code in Algorithm 1, more cases in AppendixB.5.

## 2.4 THE CNMP$_+$ STRATEGY

The CNMP strategy efficiently maintains the proportion of relevant rules, although it still exists the relevant rule loss issue. As depicted in Figure 2, the entities $e_1$ and $e_3$ on the 2-hop enclosing

---

**Algorithm 1** Pseudo code for CNMP strategy

---

1: **Input** the target prediction link $(u, r_t, v)$ and the k-hop enclosing subgraph $\mathcal{G}$.
2: Define the $k$-hop neighbor of node $u$ as $\mathcal{N}_k(u)$ and the distance in the original knowledge graph between node $i$ and node $u$ as $d(i, u)$.
3: Compute the intersection of $k$-hop neighbor of the target nodes $u$ and $v$ to get the common neighbor set, $S = \mathcal{N}_k(u) \cap \mathcal{N}_k(v)$.
4: **for** Isolated Node $i$ in $S$ **do**
5:   Do the **Label Procedure** for nodes $i, u$ and $i, v$ respectively to get the labeled relevant nodes and relations.
6: **end for**
7: **Label Procedure**
8: Compute $d(i, u)$ and $d(i, v)$ for node $i$ and the target nodes $u$ and $v$.
9: Label the isolated common neighbor $i$ with $d(i, u)$ and $d(i, v)$.
10: Label the relation between nodes $i$ and $j$ with $r^{'} = r \oplus (d(i, u), d(i, v))$, where $\oplus$ denotes the concatenation of two vectors and $r$ is the same as the isolated common neighbor node $i$ to its adjacent neighboring node in the original knowledge graph.
11: Update the Labeled node $i$ in the intersection set $S$.
12: **End Label Procedure**
13: Assigning the results $S$ of S to CNMP_Computational Graph
14: **return** CNMP_Computational Graph

---

**Algorithm 2** Pseudo code for CNMP$_+$ strategy

---

1: **Input** the target prediction link $(u, r_t, v)$ and the k-hop enclosing subgraph $\mathcal{G}$.
2: Define $\mathcal{N}_i(u)$ as $i$ hop neighbor of the node $u$, $\mathcal{B}_i(u)$ as the $i^{th}$ hop neighbor of $u$ and $\mathcal{N}_0(u) = u$ as the zero hop neighbor of $u$.
3: **for** $i = 1, 2, ..., k$ **do**
4:   $\mathcal{B}_i(u) = \mathcal{N}_i(u) \setminus \mathcal{N}_{i-1}(u)$, $\mathcal{B}_i(v) = \mathcal{N}_i(v) \setminus \mathcal{N}_{i-1}(v)$
5: **end for**
6: Do the **Distilled Procedure** for node $u$ and $v$ to get distilled relevant neighbors $\mathcal{N}_i(u)$ and $\mathcal{N}_i(u)$, respectively.
7: **Distilled Procedure**
8: **for** $i = k - 1, .., 1$ **do**
9:   **for** node in $\mathcal{N}_i(u)$ **do**
10:     **if** $\mathcal{N}_1(node) \cap \mathcal{B}_{i+1}(u) = \phi$ **then**
11:       $\mathcal{N}_i(u) = \mathcal{N}_i(u) \setminus node$
12:     **end if**
13:   **end for**
14: **end for**
15: **End Distilled Procedure**
16: Compute the $k$ hop common neighbor sets, $Int\_k = \mathcal{N}_k(u) \bigcap \mathcal{N}_k(v)$
17: Compute the $i^{th}$ hop $(i = 0, 1 \cdots, k)$ distilled union neighbors, $Distilled\_k = \bigcup_{i=0}^{k-1} \mathcal{N}_i(u) \bigcup \bigcup_{i=0}^{k-1} \mathcal{N}_i(v)$
18: Compute the CNMP$_+$_Computational_Graph $= Int\_k \bigcup Distilled\_k$
19: **return** CNMP$_+$_Computational_Graph

---

subgraph are omitted during message passing. Thus, we introduce the CNMP$_+$ strategy, which aims to comprehensively preserve the rules related to the target head entity $u$ and the target tail entity $v$. Specifically, CNMP$_+$ maintains a list of common neighbors at different distances from the target nodes. By employing a probing strategy, it starts from isolated common neighbors and orderly explores other common neighbors closer to the target nodes. During this process, only the nodes most likely to be found along the reasoning path are retained, while others are discarded, iteratively restoring the reasoning path. CNMP$_+$ not only preserves all relevant rules by achieving more complete message passing but also avoids redundant information since each node and relation is indispensable on the reasoning path. As shown by Figures 2, the CNMP$_+$ strategy adequately preserves relevant rules while eliminating irrelevant rules. Additionally, we provide the pseudo code

of the algorithm in Algorithm 2 and more cases in AppendixB.5 to describe this process in detail for clarity.

# 3 Experiments and Analysis

This section is organized as follows. **First**, we demonstrate the efficacy of our CNMP models on different inductive benchmarks in Section 3.3. **Second**, we perform experiments on large inductive link prediction benchmarks to demonstrate the scalability of our CNMP models in Section 3.4. **Third**, following RMPI (Geng et al., 2022), we conduct the fully inductive link prediction to demonstrate the generalization ability of CNMP models in Section 3.5. **Finally**, we conduct the ablation study in Appendix B.7 and further experiments in Appendix B.8 to provide more insights to our CNMP models. Moreover, we also provide further statistical analysis in Appendix C for comparison with Section 2.2.

## 3.1 Datasets

We use the benchmarks for link prediction introduced in GraIL (Teru et al., 2020), which are derived from WN18RR (Toutanova & Chen, 2015), FB15k-237 (Dettmers et al., 2018), and NELL-995 (Xiong et al., 2017). For inductive link prediction, the training and testing set should have no overlapping entities. Each knowledge graph of WN18RR, FB15k-237, and NELL-995 induces four versions with increasing sizes. Please refer to the Appendix B for details.

## 3.2 The baseline model

For convenience, we refer to RCN with CNMP strategy applied as CNMP-base model, and RCN with CNMP$_+$ strategy as CNMP$_+$-base model. Further incorporating RGCN, we introduce CNMP model and CNMP$_+$ model respectively. Please refer to Appendix D.4 and D.5 for details.

## 3.3 Inductive Link Prediction

We evaluate the models on both classification and ranking metrics. For both metrics, we compare our method to existing state-of-the-art methods, including Neural LP (Yang et al., 2017), DRUM (Sadeghian et al., 2019), RuleN (Meilicke et al., 2018), GraIL (Teru et al., 2020), CoMPILE (Mai et al., 2021), RMPI (Geng et al., 2022), ConGLR(Lin et al., 2022), NBFNet (Zhu et al., 2021), SNRI (Xu et al., 2022), and TACT(Chen et al., 2021). Nearly all the existing state-of-the-art methods are based on rule learning or enclosing graphs. They have advantages such as rich logical information and good scalability, but may lose key entities and relations during the extraction of enclosing subgraphs, leading to many disconnected reasoning paths, which severely hinders message passing and consequently impacts overall reasoning performance. Comparing our approach with these methods can demonstrate the superiority of our approach.

### 3.3.1 Classification metric

We use the area under the precision-recall curve (AUC-PR) as the classification metric following GraIL (Teru et al., 2020). We substitute either the head or the tail entity of each test triple with a randomly selected entity to generate the corresponding negative triples through sampling. Then we score the positive triples with an equal number of negative triples to calculate AUC-PR following GraIL. To make the results more reliable, we run each experiment five times with different random seeds and report the mean results.

From the AUC-PR results in Table 3, we observe that on the twelve versions of the three datasets,models based on CNMP strategies have reached the optimum in all AUC-PR values. Specifically, CNMP models outperform rule-based baselines, including Neural LP, DRUM, and RuleN by a significant margin. Compared with the subgraph-based models GraIL, CoMPILE, TACT, RMPI, ConGLR, and SNRI, the best performance CNMP models can achieve average AUC-PR improvements of 7.58%, 7.27%, 9.98%; 1.54%, 6.27%, 10.47% ; 4.13%, 3.03%, 4.04%; 5.90%, 6.38%, 9.88%; 1.11%, 5.87%, 8.27%; and 0.98%, 6.85% , 6.35% on three datasets, respectively, which demonstrates the superiority of CNMP. And it also outperforms NBFNet with average AUC-PR improvements of

Table 3: AUC-PR results on inductive benchmarks. The results of Neural LP, DRUM, RuleN, GraIL, CoMPILE, ConGLR are taken from the ConGLR (Lin et al., 2022). The results of RMPI, SNRI and TACT are all from the corresponding published papers.

| | WN18RR | | | | FB15k-237 | | | | NELL-995 | | | |
|---|---|---|---|---|---|---|---|---|---|---|---|---|
| | v1 | v2 | v3 | v4 | v1 | v2 | v3 | v4 | v1 | v2 | v3 | v4 |
| Neural LP | 86.02 | 83.78 | 62.90 | 82.06 | 69.64 | 76.55 | 73.95 | 75.74 | 64.66 | 83.61 | 87.58 | 85.69 |
| DRUM | 86.02 | 84.05 | 63.20 | 82.06 | 69.71 | 76.44 | 74.03 | 76.20 | 59.86 | 83.99 | 87.71 | 85.94 |
| RuleN | 90.26 | 89.01 | 76.46 | 85.75 | 75.24 | 88.70 | 91.24 | 91.79 | 84.99 | 88.40 | 87.20 | 80.52 |
| GraIL | 94.32 | 94.18 | 85.80 | 92.72 | 84.69 | 90.57 | 91.68 | 94.46 | 86.05 | 92.62 | 93.34 | 87.50 |
| CoMPILE | 98.23 | 99.56 | 93.60 | 99.80 | 85.50 | 91.86 | 93.12 | 94.90 | 80.16 | 95.88 | 96.08 | 85.48 |
| RMPI | 95.05 | 95.48 | 88.35 | 94.87 | 85.90 | 92.96 | 92.72 | 93.33 | 77.89 | 94.31 | 95.89 | 91.77 |
| ConGLR | 99.58 | 99.67 | 93.78 | 99.88 | 85.68 | 92.32 | 93.91 | 95.05 | 86.48 | 95.22 | 96.16 | 88.46 |
| NBFNet | 98.39 | 98.96 | 94.37 | 99.12 | 93.06 | 96.88 | 97.05 | 97.83 | 98.30 | 98.22 | 97.89 | 98.22 |
| SNRI | 99.10 | 99.92 | 94.90 | 99.61 | 86.69 | 91.77 | 91.22 | 93.37 | 90.54 | 93.31 | 98.37 | 91.79 |
| TACT | 96.15 | 97.95 | 90.58 | 96.15 | 88.73 | 94.20 | 97.10 | 98.30 | 94.87 | 96.58 | 95.70 | 96.12 |
| CNMP-base | 98.90 | 97.94 | 91.23 | 97.85 | 92.12 | 96.87 | 98.08 | 98.34 | 99.60 | 99.27 | 99.07 | 98.54 |
| CNMP | 99.27 | 98.41 | 93.90 | 99.27 | **93.97** | **97.40** | 98.83 | 99.39 | 99.69 | 99.17 | 99.30 | 99.07 |
| CNMP$_+$-base | **99.73** | **99.94** | 95.26 | 97.92 | 91.73 | 96.67 | 98.80 | 98.10 | 99.30 | 99.06 | 99.09 | 98.64 |
| CNMP$_+$ | 99.14 | 99.77 | **97.75** | **99.94** | 92.54 | 96.45 | **99.66** | **99.43** | **99.95** | **99.92** | **99.98** | **99.56** |

1.63%; 1.41%; 1.70% on all three datasets, respectively. These results highlight the significance of our new messaging strategy, demonstrating remarkable performance based on simple backbones.

### 3.3.2 RANKING METRIC

We assess the ranking of each test triple relative to 50 randomly selected negative triples. Specifically, when evaluating a relation prediction of the form $(u, r_t, ?)$ or $(?, r_t, v)$ in the testing dataset, we rank the true triples $(u, r_t, v)$ against all other candidate negative triples. Following TransE (Bordes et al., 2013), we use the filtered setting, which excludes any pre-existing valid triples from consideration during ranking. We choose Hits at N (H@N) as the evaluation metric. For a fair comparison, we set the negative sampling rate as 1 for all compared baselines in the training stage. Following GraIL(Teru et al., 2020), to make the results more reliable, we run each experiment five times with different random seeds and report mean results.

Table 4: HITS@10 results on inductive benchmarks datasets. The results of Neural LP, DRUM, RuleN, GraIL, CoMPILE, ConGLR are taken from the ConGLR (Lin et al., 2022). The results of RMPI, SNRI and TACT are all from the corresponding published papers.

| | WN18RR | | | | FB15k-237 | | | | NELL-995 | | | |
|---|---|---|---|---|---|---|---|---|---|---|---|---|
| | v1 | v2 | v3 | v4 | v1 | v2 | v3 | v4 | v1 | v2 | v3 | v4 |
| Neural LP | 74.37 | 68.93 | 46.18 | 67.13 | 52.92 | 58.94 | 52.90 | 55.88 | 40.78 | 78.73 | 82.71 | 80.58 |
| DRUM | 74.73 | 68.93 | 46.18 | 67.13 | 52.92 | 58.73 | 52.90 | 55.88 | 19.42 | 78.55 | 82.71 | 80.58 |
| RuleN | 80.85 | 78.23 | 53.39 | 71.59 | 49.76 | 77.82 | 87.69 | 85.60 | 53.50 | 81.75 | 77.26 | 61.35 |
| GraIL | 82.45 | 78.68 | 58.43 | 73.41 | 64.15 | 81.80 | 82.83 | 89.29 | 59.50 | 93.25 | 91.41 | 73.19 |
| CoMPILE | 83.60 | 79.82 | 60.69 | 75.49 | 67.66 | 82.98 | 84.67 | 87.44 | 58.38 | 93.87 | 92.77 | 75.19 |
| RMPI | 87.77 | 82.43 | 73.14 | 81.42 | 71.71 | 83.37 | 86.01 | 88.69 | 60.50 | 93.49 | 95.30 | 66.42 |
| ConGLR | 85.64 | 92.93 | 70.74 | 92.90 | 68.29 | 85.98 | 88.61 | 89.31 | 81.07 | 94.92 | 94.36 | 81.61 |
| NBFNet | 91.90 | 90.53 | 89.35 | 88.60 | 81.30 | **90.62** | **95.00** | 90.41 | 63.50 | 94.96 | 93.97 | 82.77 |
| SNRI | 87.23 | 83.10 | 67.31 | 83.32 | 71.79 | 86.50 | 89.59 | 89.39 | 61.50 | 91.37 | 93.31 | 81.19 |
| TACT | 81.69 | 80.06 | 62.32 | 74.69 | 65.48 | 84.25 | 85.62 | 88.04 | 58.00 | 91.17 | 90.72 | 73.42 |
| CNMP-base | 90.08 | **92.98** | **89.67** | 90.69 | 78.05 | 87.17 | 88.32 | 87.15 | 77.00 | 94.01 | 93.30 | 82.79 |
| CNMP | 91.09 | 91.40 | 85.45 | 88.59 | **82.01** | 86.19 | 86.53 | 89.96 | 79.00 | 93.30 | **95.49** | **84.33** |
| CNMP$_+$-base | 93.22 | 88.73 | 79.84 | 90.71 | 79.93 | 88.52 | 89.62 | 90.84 | **82.00** | 93.71 | 93.11 | 79.09 |
| CNMP$_+$ | **95.12** | 90.46 | 82.02 | **93.37** | 75.93 | 86.83 | 85.59 | **91.17** | 81.50 | **95.27** | 94.28 | 77.19 |

Table 4 shows the HITS@10 results for WN18RR, FB15k-237, and NELL-995 across versions 1 to 4. As we can see, CNMP models significantly outperform rule-based methods Neural LP, DRUM, and RuleN; subgraph-based methods GraIL, TACT, CoMPILE, and SNRI in all datasets by a significant margin. In this scenario, CNMP models achieve a maximum of 27.35% (on WN18RR v3) and 15.88% (on WN18RR v4). Compared with the subgraph-based method GraIL, CoMPILE, TACT,

Table 5: Scalability evaluation of GraIL, NBFNet, TACT, and CNMP on ILPC small and large datasets. We choose the Mean Reciprocal Rank (MRR) and HITS@10 for evaluation metrics. OOM denotes the out-of-memory issue.

| | ILPC-small | | ILPC-large | |
|---|---|---|---|---|
| | MRR | HITS@10 | MRR | HITS@10 |
| GraIL | 46.15 | 73.41 | 45.66 | 66.56 |
| NBFNet | OOM | OOM | OOM | OOM |
| TACT | 50.22 | 81.32 | 52.39 | 75.38 |
| CNMP | **58.31** | **88.74** | **58.09** | **81.74** |

RMPI, ConGLR, and SNRI, the best performance CNMP models can achieve average HITS@10 improvements of 18.82%, 8.31%, 4.37%; 17.16%, 7.15%, 3.65% ; 17.38%, 6.29%, 3.62%; 11.59%, 5.38%, 10.34%; 7.23%, 4.78%, 1.28%; and 11.82%, 3.51% , 7.43% on three datasets, respectively, which demonstrates the superiority of CNMP models. And it also outperforms NBFNet with average HITS@10 improvements of 2.69% and 5.47% on WN18RR and NELL-995 datasets respectively.

As the aforementioned Table 1, the Incomplete_Ratio serves as an indicator of the relevant rule loss degree incurred by the enclosing subgraph method. The smaller the Incomplete_Ratio value is, the more relevant rule loss is caused by the enclosing subgraph method. FB15k-237 exhibits the highest Incomplete_Ratio values in both training and testing datasets, thus the proposed CNMP method exhibit relatively modest improvements compared with the datasets WN18RR and NELL995.

## 3.4 RESULTS ON LARGE KNOWLEDGE GRAPHS

CNMP is an improved message passing framework and CNMP strategy operates by restoring path connectivity while $CNMP_+$ strategy operates by iteratively reconstructing the accurate and complete paths, so it is not affected by the whole graph scale. This characteristic suggests the significant scalability potential of CNMP when applied to large knowledge graphs. Therefore, we conduct an evaluation involving GraIL, NBFNet, TACT, and CNMP on the ILPC datasets (Galkin et al., 2022). The ILPC-small and ILPC-large datasets (Galkin et al., 2022) are sourced from Wikidata (Vrandečić & Krötzsch, 2014), which stands as the largest open source KG. Both datasets represent a real-world KG used in many practical downstream tasks and are larger than existing benchmarks, which is challenging for existing inductive baselines. These datasets feature a higher volume of relations, entities, and triples as shown in Table 8.

As demonstrated in Table 5, NBFNet (Zhu et al., 2021) encounters serious scalability issues in large KGs. Specifically, NBFNet reasons on the whole KG, resulting in the out of memory issue. For both ILPC-small and ILPC-large datasets, GraIL and TACT both attains suboptimal results. CNMP models consistently attains optimal results across both ILPC-small and ILPC-large datasets, which also demonstrates the effectiveness of our CNMP strategies.

## 3.5 FULLY INDUCTIVE LINK PREDICTION RESULTS

To enable a comprehensive evaluation of fully inductive KG completion, especially when dealing with previously unseen relations in the testing graph, we follow RMPI (Geng et al., 2022) to re-combine the existing 12 inductive benchmarks. Ultimately, we construct four datasets in which the entities and relations are unseen in the training set; for details on the data construction process, please refer to Appendix B.6. We further validate the effectiveness of CNMP through this fully inductive link prediction experiment.As shown in Table 6, CNMP models outperform RMPI, NBFNet, GraIL, and TACT in all fully inductive benchmarks consistently. The results further demonstrate the effectiveness of CNMP strategies across different scenarios.

Table 6: The results of both classification and ranking metrics on fully inductive benchmarks datasets. The results of RMPI and TACT are taken from the paper RMPI (Geng et al., 2022).

| Methods | NELL-995.v1.v3 | | | NELL-995.v2.v3 | | | NELL-995.v4.v3 | | | FB15k-237.v1.v4 | | |
|---|---|---|---|---|---|---|---|---|---|---|---|---|
| | AUC-PR | MRR | HITS@10 | AUC-PR | MRR | HITS@10 | AUC-PR | MRR | HITS@10 | AUC-PR | MRR | HITS@10 |
| RMPI | 84.86 | 59.10 | 82.12 | 91.10 | 73.90 | 88.78 | 88.20 | 70.33 | 81.20 | 88.99 | 57.77 | 80.38 |
| NBFNet | 85.39 | 56.57 | 77.32 | 86.93 | 67.32 | 88.33 | 88.31 | 68.64 | 83.71 | 92.13 | 61.89 | 88.02 |
| GraIL | 74.71 | 44.19 | 68.87 | 78.85 | 54.03 | 75.76 | 70.78 | 44.16 | 62.04 | 83.35 | 46.33 | 66.80 |
| TACT | 73.98 | 43.59 | 72.48 | 85.88 | 65.63 | 83.47 | 72.40 | 52.68 | 67.95 | 88.76 | 56.81 | 79.71 |
| CNMP-base | 85.88 | 57.55 | 83.86 | 93.19 | 68.33 | 89.94 | 86.19 | 65.58 | 84.96 | 90.94 | 59.11 | 81.32 |
| CNMP | 86.89 | 62.38 | 85.66 | 92.82 | 70.19 | 88.35 | 86.37 | 64.35 | 85.21 | 90.45 | 59.89 | 85.32 |
| $CNMP_+$-base | 84.52 | 59.88 | 84.03 | 93.38 | 67.26 | 90.13 | 89.22 | 68.37 | 84.02 | 94.48 | **62.33** | **88.10** |
| $CNMP_+$ | **88.70** | **66.28** | **89.57** | **94.21** | **75.85** | **91.85** | **92.37** | **70.58** | **88.06** | **95.31** | 61.05 | 86.79 |

# 4 RELATED WORK

## 4.1 TRANSDUCTIVE LINK PREDICTION

Link prediction is the problem of predicting the existence of a link between two nodes within a network(Liben-Nowell & Kleinberg, 2003). Following the success of word embeddings in language modeling Mikolov et al. (2013); Pennington et al. (2014), various transductive link prediction models have been developed based on entity and relation embeddings, including TransE(Bordes et al., 2013), TransR(Lin et al., 2015a), ComplEx(Trouillon et al., 2016) and Distmult(Yang et al., 2015). However, these methods can only handle the entities seen during the training stage(Sadeghian et al., 2019).

## 4.2 INDUCTIVE LINK PREDICTION

Inductive link prediction can predict missing relations between unseen entities during training. Rule-based methods and GNN-based methods are two mainstream approaches for inductive link prediction. Rule-based approaches (Cohen, 2016; Yang et al., 2017; Sadeghian et al., 2019) aim to discover logical rules from frequent co-occurrence patterns of relations, and these rules are entity-independent, making them inherently inductive. While traditional rule-based methods face scalability and expressive power issues with large knowledge graphs, recent methods like Neural LP (Yang et al., 2017) and DRUM (Sadeghian et al., 2019) improve this by using differentiable approaches and low-rank matrix approximation, respectively. GNN-based methods, such as GraIL (Teru et al., 2020) and CoMPILE (Mai et al., 2021) , enhance link prediction by reasoning over subgraphs, but they overlook semantic correlations between relations. To address this, approaches like SNRI (Xu et al., 2022) and ConGLR (Lin et al., 2022) improve the representation of neighboring relations, while NBFNet (Zhu et al., 2021) introduces a neural Bellman-Ford network for link prediction. More details about inductive link prediction can be found in Appendix D.

# 5 CONCLUSION

In this paper, we propose a novel Common Neighbor Induced Message Passing framework (CNMP), which effectively ensures message passing even when reasoning paths in subgraphs are disconnected, and to avoid introducing irrelevant information. Specifically, for disconnected reasoning paths, we propose CNMP strategy to efficiently restore path connectivity, and further introduce $CNMP_+$ strategy to iteratively reconstruct the accurate and complete paths. To demonstrate the effectiveness of our method, we combine the CNMP strategies with classical GNN-based methods in inductive link prediction as the CNMP models. Experiments demonstrate that CNMP models accurately extract key information from subgraphs based on a new messaging mechanism, leading to superior performance over existing state-of-the-art methods on inductive link prediction. Moreover, experiments under large KGs and fully inductive settings demonstrate a superior scalability and generalization ability of CNMP models.

Regarding the limitations of CNMP method, although it achieves remarkable performance in the inductive link prediction task with classical RCN and RGCN, more new GNN-based methods can be explored in future works. Moreover, CNMP focuses on inductive link prediction tasks, and its effectiveness can be verified on various forms of tasks in the future.

## 6 ETHICS STATEMENT

This paper introduces the Common Neighbor Induced Message Passing (CNMP) framework for inductive link prediction in knowledge graphs. Our research adheres to ethical standards, as it does not involve human subjects or sensitive data. The methods discussed focus on enhancing message passing in knowledge graphs without any harmful applications identified. We emphasize responsible use of our findings, particularly in high-stakes scenarios, to avoid potential misinterpretations. No conflicts of interest were found, and all experiments comply with relevant ethical standards

## 7 REPRODUCIBILITY STATEMENT

In this study, to ensure the reproducibility of our approach, we provide key information from the main text and Appendix as follows.

1. **Algorithm.** We provide the architecture and pseudo code of our approach CNMP and CNMP$_+$ in Section 2.3 and Section 2.4 respectively. We also provide the implementation details of CNMP and CNMP$_+$ in Appendices B. See Appendix B.4 for the hyperparameters of CNMP and CNMP$_+$. Moreover, we are committed to providing the source code of our approach, if accepted.

2. **Experimental Details.** We provide detailed experiment settings in Section 3 and Appendices B.1 and B.2.

3. **Theoretical Proofs.** We provide all proofs in Appendix A.

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

# A   PROOF OF COMMON NEIGHBORS IN A REASONING PATH

## A.1   PART 1: SINGLE-HOP NEIGHBORS

Let $G = (V, E)$ be a graph with vertex set $V$ and edge set $E$. For any vertex $x$, its *single-hop neighborhood*, denoted $N(x)$, is defined as the set of vertices directly connected to $x$, i.e., $N(x) = \{y \in V | (x, y) \in E \text{ or } (y, x) \in E\}$. A *reasoning path* is a sequence of vertices $(x_1, x_2, \ldots, x_n)$ where $x_1 = u$, $x_n = v$, and $(x_i, x_{i+1}) \in E$ for all $1 \leq i < n$, denoted $P_{uv}$.

**Theorem:** If two vertices $u, v \in V$ have a common single-hop neighbor $c$, then there exists a reasoning path $P_{uv}$ from $u$ to $v$ that includes $c$.

**Proof of Theorem:** Given $u, v \in V$ and a common single-hop neighbor $c \in N(u) \cap N(v)$, by definition, $(u, c) \in E$ and $(c, v) \in E$. The sequence of vertices $(u, c, v)$ forms a valid reasoning path $P_{uv}$, since it starts with $u$, ends with $v$, and each consecutive pair of vertices is connected by an edge in $E$. ∎

## A.2   PART 2: MULTI-HOP NEIGHBORS

For any vertex $x \in V$ and a positive integer $k$, the *k-hop neighborhood* of $x$, denoted by $N_k(x)$, is defined as the set of vertices that can be reached from $x$ by traversing exactly $k$ edges. Formally, $N_k(x)$ includes vertex $y$ if there exists a sequence of vertices $(x = x_0, x_1, \ldots, x_k = y)$ such that $(x_{i-1}, x_i) \in E$ for all $1 \leq i \leq k$.

**Theorem:** If two vertices $u, v \in V$ have a common multi-hop neighbor $c$, then there exists a reasoning path $P_{uv}$ from $u$ to $v$ that includes $c$.

**Proof of Theorem:** Let $u, v \in V$ be two vertices with a common multi-hop neighbor $c \in V$. This implies that for some positive integer $k$, $c \in N_k(u) \cap N_k(v)$. By the definition of $N_k$, there exist paths $P_{uc}$ from $u$ to $c$ and $P_{cv}$ from $c$ to $v$, each consisting of $k$ edges.

To construct a reasoning path from $u$ to $v$ that includes the common multi-hop neighbor $c$, we concatenate the paths $P_{uc}$ and $P_{cv}$ where $c$ is the common vertex between the two paths. Formally, we can write:

$$P_{uv} = P_{uc} \oplus P_{cv} \tag{1}$$

where $\oplus$ denotes the concatenation of the paths such that the common vertex $c$ appears only once in the concatenated path $P_{uv}$. Explicitly, if

$$P_{uc} = (u, \ldots, c)$$
$$P_{cv} = (c, \ldots, v)$$

then the concatenated path $P_{uv}$ is given by:

$$P_{uv} = (u, \ldots, c, \ldots, v)$$

This results in a reasoning path that starts at $u$, passes through $c$, and ends at $v$, demonstrating that there exists such a path $P_{uv}$ that includes $c$. ∎

# B   IMPLEMENTATION DETAILS OF CNMP

## B.1   DATASETS

Please refer to table 7 for statistics of inductive benchmarks.

Table 7: Statistics of inductive benchmarks. We employ #E and #R, along with #TR, to represent the quantities of entities, relations, and triples, respectively.

| | | WN18RR | | | FB15k-237 | | | NELL-995 | | |
|---|---|---|---|---|---|---|---|---|---|---|
| | | #R | #E | #TR | #R | #E | #TR | #R | #E | #TR |
| v1 | train | 9 | 2746 | 6678 | 180 | 1594 | 5226 | 14 | 3130 | 5540 |
| | test | 8 | 922 | 1991 | 142 | 1093 | 2404 | 14 | 225 | 1034 |
| v2 | train | 10 | 6954 | 18968 | 200 | 2608 | 12085 | 88 | 2564 | 10109 |
| | test | 10 | 2757 | 4863 | 172 | 1660 | 5092 | 79 | 2086 | 5521 |
| v3 | train | 11 | 12078 | 32150 | 215 | 3668 | 22394 | 142 | 4647 | 20117 |
| | test | 11 | 5084 | 7470 | 183 | 2501 | 9137 | 122 | 3566 | 9668 |
| v4 | train | 9 | 3861 | 9842 | 219 | 4707 | 33916 | 76 | 2092 | 9289 |
| | test | 9 | 7084 | 15157 | 200 | 3051 | 14554 | 61 | 2795 | 8520 |

## B.2 TRAINING IMPLEMENTATION

We conduct all the experiments on an Nvidia GeForce RTX 3090 GPU and an Intel(R) Xeon(R) Gold 6246R CPU @ 3.40GHz. We use PyTorch(Paszke et al., 2019) and DGL(Wang et al., 2019) to implement all our models based on CNMP strategies. For optimization, we use Adam optimizer (Kingma & Ba, 2015) with batch size 16 and an initial learning rate of $1 \times 10^{-2}$. We divide the learning rate by 5 when the validation loss does not improve for 5 epochs. We use the marginal ranking loss to construct our CNMP models. The maximum training epoch of CNMP models is set to 20. When training our CNMP-base model, we use an initial learning rate of $5 \times 10^{-3}$ with the same learning rate scheduler as CNMP models but just train maximum 15 epochs. We will release our code once the paper is accepted to be published.

## B.3 MODEL FRAMEWORK

First, we apply a graph extractor for each target link to get enclosing subgraphs and corresponding RCGs. Then, we apply a two-layer RGCN with CNMP strategy to reason on enclosing and a two-layer RCN with CNMP strategy to reason on their corresponding relation correlation graphs. Finally, for the CNMP models, we apply the combination of RGCN's embedding vectors and relation correlation RCN's embedding vectors into a single layer perceptron to produce a plausibility score of the target prediction relation link and other negative candidate sample links for both classification and rank tasks. Note that the CNMP-base model is only based on RCN.

## B.4 HYPERPARAMETERS

For all the experiments, we set the training and validation batch size for branching models to be 16. We conduct grid search to obtain optimal hyperparameters, where we search dropout rate in $\{0, 0.1, 0.2\}$, edge-dropout rate in $\{0, 0.3, 0.5\}$ and margins in the loss function in $\{8, 10, 12, 16\}$. The regularization coefficient of GNN weights is set to 0.01. Configuration for the best performance of each dataset is given within the code.

Table 8: Statistics for ILPC-small and ILPC-large datasets. The symbols #E, #R, and #TR denote the number of edges, relations, and triples, respectively.

| | ILPC-S | | | ILPC-L | | |
|---|---|---|---|---|---|---|
| Split | #E | #R | #TR | #E | #R | #TR |
| Training | 10230 | 96 | 78616 | 46626 | 130 | 202446 |
| Inference | 6653 | 96 | 20960 | 29246 | 130 | 77044 |
| Inference (val.) | 6653 | 96 | 2908 | 29246 | 130 | 10179 |
| Inference (test.) | 6653 | 96 | 2902 | 29246 | 130 | 10184 |
| Total | 16883 | 96 | 108280 | 75872 | 130 | 310045 |

### B.5 MORE CASE FOR CNMP

In this section, we present a comparative analysis of CNMP strategies. Figure 3 contrasts the 2-hop enclosing subgraph with the CNMP strategy and the enhanced CNMP$_+$ strategy during the message-passing process within an identical original knowledge graph. Figure 4 expands on this analysis by comparing three distinct approaches: the 3-hop enclosing subgraph method, the 2-hop enclosing subgraph with the CNMP strategy, and the CNMP$_+$ strategy.

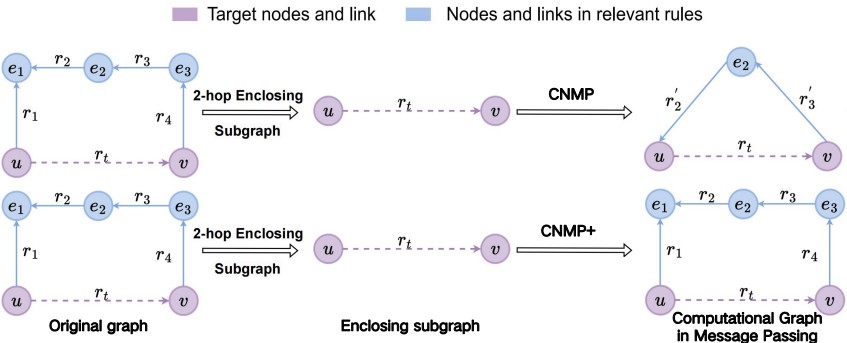

Figure 3: A comparative analysis between the 2-hop enclosing subgraph with CNMP strategy and CNMP$_+$ strategy within the same original knowledge graph during message passing.

### B.6 THE DATASET FOR FULLY INDUCTIVE LINK PREDICTION

Specifically, we retain the training graph of each original benchmark while replacing the testing graph with a newly mixed benchmark that includes a more extensive set of relations. For instance, for the second benchmark derived from NELL-995 (referred to as NELL-995.v2), comprising 88 relations, we merge it with the testing graph from NELL-995.v3, containing a total of 122 relations, 51 of which are not part of NELL-995.v2. This results in the creation of a new benchmark denominated as NELL-995.v2.v3.

These newly constructed datasets are denoted as "XXX.v$i$.v$j$", where "XXX" is the source transductive dataset, "i" is the index indicating the version of the inductive benchmark from which the training graph is sourced, and "j" is the version identifier for the origin of the testing graph. Notably, for each dataset, we filter the testing graph to ensure that none of its entities are present in the corresponding training graph. We finally get four new datasets for evaluation, as shown in Table 9.

### B.7 ABLATION STUDIES

We introduce a variety of ablation experiments to further enhance the experimental analysis. Specifically, we conduct ablation studies based on RCN, RGCN, and RCN+RGCN in AUC-PR and HITS@10 metrics. As shown in Tables 11 and 12, the CNMP strategy brings significant improvements in various settings.

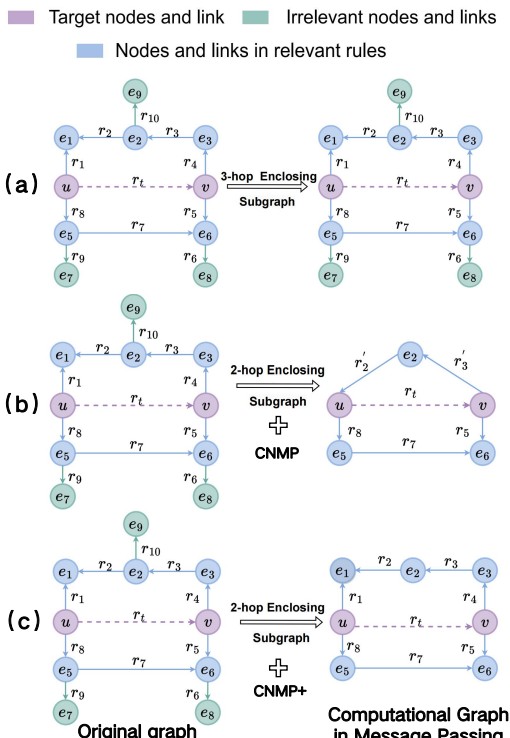

Figure 4: A comparative analysis among the 3-hop enclosing subgraph method, the 2-hop enclosing subgraph with CNMP strategy and CNMP$_+$ strategy within the same original knowledge graph during message passing.

Table 9: Statistics of the fully inductive benchmarks. "train" and "test" represent the training and testing graphs, respectively. "#R/E/TR" denotes the counts of relations, entities, and triples, respectively. The numbers in the brackets are the numbers of unseen relations.

|  | NELL-995.v1.v3 | | | NELL-995.v2.v3 | | |
|---|---|---|---|---|---|---|
|  | #R | #E | #TR | #R | #E | #TR |
| train | 14 | 3103 | 5540 | 88 | 2564 | 10109 |
| test | 106 (98) | 2271 | 5550 | 116 (49) | 2803 | 6749 |
|  | NELL-995.v4.v3 | | | FB15k-239.v1.v4 | | |
|  | #R | #E | #TR | #R | #E | #TR |
| train | 76 | 2092 | 9289 | 180 | 1594 | 5226 |
| test | 110 (53) | 3140 | 8308 | 200 (26) | 3051 | 14554 |

Table 10: Comparison between the 2-hop enclosing subgraph with CNMP, 2-hop CNMP$_+$, and 3-hop enclosing subgraph in the triple classification task.

|  | WN18RR | | | | FB15k-237 | | | | NELL-995 | | | |
|---|---|---|---|---|---|---|---|---|---|---|---|---|
|  | v1 | v2 | v3 | v4 | v1 | v2 | v3 | v4 | v1 | v2 | v3 | v4 |
| 2-hop ES + CNMP | **99.27** | 98.41 | 93.90 | 99.27 | **93.97** | **97.40** | 98.83 | 99.39 | 99.69 | 99.17 | 99.30 | 99.07 |
| 2-hop ES + CNMP$_+$ | 99.14 | **99.77** | **97.75** | **99.94** | 92.54 | 96.45 | **99.66** | **99.43** | **99.95** | **99.92** | **99.98** | **99.56** |
| 3-hop ES | 97.79 | 96.43 | 88.15 | 81.57 | 88.34 | 94.42 | 97.16 | 98.16 | 93.95 | 95.97 | 93.83 | 94.76 |

Table 11: Ablation study based on RGCN, RCN, and RCN+RGCN in triple classification task

|  | WN18RR | | | | FB15k-237 | | | | NELL-995 | | | |
|---|---|---|---|---|---|---|---|---|---|---|---|---|
| AUC-PR | v1 | v2 | v3 | v4 | v1 | v2 | v3 | v4 | v1 | v2 | v3 | v4 |
| RGCN | 98.11 | 97.11 | 88.34 | 97.25 | 87.36 | 94.31 | 97.42 | 98.09 | 94.00 | 94.44 | 93.98 | 94.93 |
| RGCN + CNMP | 98.90 | 97.94 | 91.23 | 97.85 | **92.12** | **96.87** | 98.08 | **98.34** | **99.60** | **99.27** | 99.07 | 98.54 |
| RGCN + CNMP$_+$ | **99.73** | **99.94** | **95.26** | **97.92** | 91.73 | 96.67 | **98.80** | 98.10 | 99.30 | 99.06 | **99.09** | **98.64** |
| RCN | 94.32 | 94.18 | 85.80 | 92.72 | 84.69 | 90.57 | 91.68 | 94.46 | 86.05 | 92.62 | 93.34 | 87.50 |
| RCN + CNMP | 96.74 | 95.42 | 89.37 | 94.59 | **89.26** | 93.27 | 91.28 | **95.58** | **92.5** | 95.97 | 92.61 | 90.46 |
| RCN + CNMP$_+$ | **97.49** | **97.28** | **92.42** | **95.58** | 88.99 | **93.95** | **92.04** | 95.08 | 88.7 | **97.82** | **93.87** | **91.22** |
| RGCN + RCN | 96.15 | 97.95 | 90.58 | 96.15 | 88.73 | 94.20 | 97.10 | 98.30 | 94.87 | 96.58 | 95.70 | 96.12 |
| RGCN + RCN + CNMP | 99.27 | 98.41 | 93.90 | 99.27 | **93.97** | **97.40** | 98.83 | 99.39 | **99.69** | 99.17 | 99.30 | 99.07 |
| RGCN + RCN + CNMP$_+$ | 99.14 | **99.77** | **97.75** | **99.94** | 92.54 | 96.45 | **99.66** | **99.43** | 99.95 | **99.92** | **99.98** | **99.56** |

Table 12: Ablation study based on RGCN, RCN, and RCN+RGCN in triple ranking task

|  | WN18RR | | | | FB15k-237 | | | | NELL-995 | | | |
|---|---|---|---|---|---|---|---|---|---|---|---|---|
| HIST@10 | v1 | v2 | v3 | v4 | v1 | v2 | v3 | v4 | v1 | v2 | v3 | v4 |
| RGCN | 81.38 | 77.64 | 58.76 | 73.47 | 64.61 | 82.72 | 86.72 | 89.71 | 58.5 | 92.16 | 91.04 | 71.33 |
| RGCN + CNMP | 90.08 | **92.98** | **89.67** | 90.69 | 78.05 | 87.17 | 88.32 | 87.15 | 77 | **94.01** | **93.3** | **82.79** |
| RGCN + CNMP$_+$ | **93.22** | 88.73 | 79.84 | **90.71** | **79.93** | **88.52** | **89.62** | **90.84** | 82 | 93.71 | 93.11 | 79.09 |
| RCN | 82.45 | 78.68 | 58.43 | 73.41 | 64.15 | 81.8 | 82.83 | 89.29 | 59.5 | 93.25 | 91.41 | 73.19 |
| RCN + CNMP | 88.14 | **89.82** | **84.23** | 86.12 | **81.93** | 85.21 | 83.56 | 90.72 | 73 | 92.59 | **93.65** | **83.82** |
| RCN + CNMP$_+$ | **91.07** | 88.16 | 73.20 | **91.03** | 74.28 | 85.19 | 83.01 | 89.5 | **78** | **93.89** | 92.41 | 77.99 |
| RGCN + RCN | 81.69 | 80.06 | 62.32 | 74.69 | 65.48 | 84.25 | 85.62 | 88.04 | 58 | 91.17 | 90.72 | 73.42 |
| RGCN + RCN + CNMP | 91.09 | **91.4** | 85.45 | 88.59 | **82.01** | 86.19 | 86.53 | 89.96 | 79 | 93.3 | **95.49** | **84.33** |
| RGCN + RCN + CNMP$_+$ | **95.12** | 90.46 | 82.02 | **93.37** | 75.93 | **86.83** | 85.59 | **91.17** | **81.5** | **95.27** | 94.28 | 77.19 |

## B.8 FURTHER EXPERIMENTS

### B.8.1 COMPARISON
#### OF THE 3-HOP ENCLOSING SUBGRAPH

As discussed in Section 2.2, the 3-hop enclosing subgraph suffers from eliminating irrelevant rules, which may cause the model to overfit to the extracted irrelevant rules within the subgraph and hinder the model performance. To further demonstrate the effectiveness of eliminating irrelevant rules of the proposed complete subgraph, we compare between these methods. As Table 10 shows, the 2-hop enclosing subgraphs with CNMP and CNMP$_+$

strategy outperform the 3-hop Enclosing subgraph consistently and significantly, which demonstrates the effectiveness of CNMP methods.

### B.8.2 RESULTS ON YAGO3-10

Table 13: The results for inductive link prediction of GraIL, TACT, and the proposed CNMP$_+$ on the dataset YAGO3-10.

|         | AUC-PR | MRR   | Hits@1 |
|---------|--------|-------|--------|
| GraIL   | 0.634  | 0.158 | 0.048  |
| TACT    | 0.915  | 0.406 | 0.140  |
| CNMP$_+$ | **0.930** | **0.471** | **0.184** |

To demonstrate the effectiveness of our proposed method on a larger knowledge graph with few relations. We conduct experiments on YAGO3-10, which is a subset of YAGO3 (Mahdisoltani et al., 2015) and contains 37 relations and 123182 entities. Table 13 shows the results for inductive link prediction of GraIL, TACT, and CNMP$_+$ on YAGO3-10. As we can see, CNMP$_+$ method outperforms GraIL and TACT by all the metrics, which demonstrates our proposed method can effectively deal with a larger knowledge graph with few relations and it also helps to improve CNMP by preserving more relevant rules.

### B.8.3 RUNNING TIME

Table 14 presents the running time, containing both training and inference stages, for GraIL, TACT, CNMP, and NBFNet. We can observe that CNMP and TACT require more time than GraIL to model the correlations between relations but the additional computational cost is insignificant compared to the performance improvement. In contrast, NBFNet has the longest running time, as it reasons on the whole graph for each target link.

Table 14: The running time of GraIL, TACT, CNMP, CNMP$_+$, and NBFNet on the version 1 of the inductive datasets. We measure these methods on the same device for a fair comparison.

|         | WN18RR(v1) | FB15k-237(v1) | NELL-995(v1) |
|---------|------------|---------------|--------------|
| GraIL   | 0.05 h     | 0.11 h        | 0.06 h       |
| TACT    | 0.07 h     | 0.13 h        | 0.09 h       |
| CNMP    | 0.08 h     | 0.15 h        | 0.10 h       |
| CNMP$_+$ | 0.11 h     | 0.18 h        | 0.12 h       |
| NBFNet  | 0.24 h     | 0.27 h        | 0.19 h       |

## C STATISTICS OF 2 AND 3 HOP ENCLOSING SUBGRAPHS WITH CNMP$_+$ STRATEGY IN BOTH TRAINING AND TESTING ILP BENCHMARKS.

As mentioned in Table 1, the enclosing subgraphs confront the empty graph issues. To provide more insights of the CNMP$_+$ computational graphs, we report the statistics of CNMP$_+$ computational graphs on inductive datasets when setting the neighbor hop $h$ to 2 and 3 in Table 15.

## D MORE DETAILS OF BACKGROUNDS

We present more background knowledge about **knowledge graph**, **link prediction**, and **graph neural network** to offer a more comprehensive understanding of our CNMP method.

Table 15: Statistics of enclosing subgraph with $CNMP_+$ strategy inductive datasets when setting the neighbor hop $h$ to 2 and 3. The values on the right of Num = 2, Num = 3, and Others denote the proportion of the corresponding type of subgraphs to the total number of extracted 2-hop enclosing subgraphs of each dataset. Num = 2 denotes that the extracted enclosing subgraph with $CNMP_+$ strategy only consists of the head entity $u$ and the tail entity $v$ with the target relation between them. Num = 3 denotes that apart from the head entity $u$ and the tail entity $v$, the extracted enclosing subgraph with $CNMP_+$ strategy only remains one other entity consisting of the path from $u$ to $v$. We sequentially enumerate the 2-hop enclosing subgraph with $CNMP_+$ strategy, 3-hop enclosing subgraph with $CNMP_+$ strategy in training dataset, 2-hop testing enclosing subgraph with $CNMP_+$ strategy, and 3-hop testing enclosing subgraph with $CNMP_+$ strategy in testing dataset, as illustrated below. We denote WN18RR, FB15k-237, and NELL-995 by wn, fb, and nl for short.

| 2hop training | wn(v1) | wn(v2) | wn(v3) | wn(v4) | fb(v1) | fb(v2) | fb(v3) | fb(v4) | nl(v1) | nl(v2) | nl(v3) | nl(v4) |
|---|---|---|---|---|---|---|---|---|---|---|---|---|
| Num=2 | 0.003 | 0.008 | 0.006 | 0.004 | 0.005 | 0.003 | 0.001 | 0.002 | 0.011 | 0.002 | 0.001 | 0.001 |
| Num=3 | 0.006 | 0.004 | 0.002 | 0.001 | 0.001 | 0.001 | 0.002 | 0.001 | 0.010 | 0.001 | 0.001 | 0.001 |
| Others | 0.991 | 0.988 | 0.992 | 0.995 | 0.994 | 0.996 | 0.997 | 0.997 | 0.979 | 0.997 | 0.998 | 0.998 |

| 3hop training | wn(v1) | wn(v2) | wn(v3) | wn(v4) | fb(v1) | fb(v2) | fb(v3) | fb(v4) | nl(v1) | nl(v2) | nl(v3) | nl(v4) |
|---|---|---|---|---|---|---|---|---|---|---|---|---|
| Num=2 | 0.002 | 0.001 | 0.006 | 0.001 | 0.001 | 0.002 | 0.001 | 0.001 | 0.009 | 0.002 | 0.002 | 0.002 |
| Num=3 | 0.004 | 0.007 | 0.001 | 0.001 | 0.001 | 0.001 | 0.002 | 0.001 | 0.008 | 0.002 | 0.002 | 0.001 |
| Others | 0.994 | 0.992 | 0.993 | 0.998 | 0.998 | 0.997 | 0.997 | 0.998 | 0.983 | 0.996 | 0.996 | 0.997 |

| 2hop testing | wn(v1) | wn(v2) | wn(v3) | wn(v4) | fb(v1) | fb(v2) | fb(v3) | fb(v4) | nl(v1) | nl(v2) | nl(v3) | nl(v4) |
|---|---|---|---|---|---|---|---|---|---|---|---|---|
| Num=2 | 0.006 | 0.012 | 0.012 | 0.005 | 0.011 | 0.007 | 0.002 | 0.001 | 0.009 | 0.001 | 0.001 | 0.003 |
| Num=3 | 0.002 | 0.007 | 0.005 | 0.004 | 0.005 | 0.001 | 0.001 | 0.001 | 0.009 | 0.002 | 0.001 | 0.002 |
| Others | 0.992 | 0.981 | 0.983 | 0.991 | 0.984 | 0.992 | 0.997 | 0.998 | 0.982 | 0.997 | 0.998 | 0.995 |

| 3hop testing | wn(v1) | wn(v2) | wn(v3) | wn(v4) | fb(v1) | fb(v2) | fb(v3) | fb(v4) | nl(v1) | nl(v2) | nl(v3) | nl(v4) |
|---|---|---|---|---|---|---|---|---|---|---|---|---|
| Num=2 | 0.001 | 0.001 | 0.003 | 0.002 | 0.007 | 0.005 | 0.002 | 0.001 | 0.001 | 0.001 | 0.001 | 0.002 |
| Num=3 | 0.001 | 0.001 | 0.001 | 0.001 | 0.003 | 0.001 | 0.001 | 0.001 | 0.001 | 0.001 | 0.001 | 0.001 |
| Others | 0.998 | 0.998 | 0.996 | 0.997 | 0.99 | 0.994 | 0.997 | 0.998 | 0.998 | 0.998 | 0.998 | 0.997 |

## D.1 KNOWLEDGE GRAPH.

Incorporating human knowledge is one of the key task for artificial intelligence (AI) (Ji et al., 2022). Knowledge representation and reasoning entails the structuring of knowledge in a manner that empowers intelligent systems to address intricate tasks (Newell et al., 1959). Knowledge graphs, as a structured representation of human knowledge, have attracted extensive research interest from both academia and industry (Dong et al., 2014; Nickel et al., 2015; Pan et al., 2023). A knowledge graph comprises entities, relations, and semantic descriptions. Entities contain both tangible real-world objects and abstract concepts, while relations denote the connections between entities. The semantic descriptions of entities and their relations encompass well-defined types and properties. The prevalent use of property graphs or attributed graphs is noteworthy, where nodes and relations possess corresponding properties or attributes (Bordes et al., 2011).

## D.2 LINK PREDICTION

Link prediction is the problem of predicting the existence of a link between two nodes within a network (Liben-Nowell & Kleinberg, 2003). Given the prevalence of networks, link prediction is widely used in numerous applications, including friend recommendation within social networks (Adamic & Adar, 2003), co-authorship prediction in citation networks (Shibata et al., 2012), movie recommendation on platforms such as Netflix (Bennett, 2007), forecasting protein interactions in biological networks (Qi et al., 2006), drug response prediction (Yang et al., 2019), metabolic network reconstruction (Lü & Zhou, 2011a), and identification of covert terrorist groups (Lü & Zhou, 2011b).

Link prediction pertains to the prediction of connections in homogeneous graphs, where nodes and links possess singular types. This constitutes the most elementary scenario, and a significant portion

of link prediction research concentrates on this configuration. In bipartite user-item networks, it is recognized as matrix completion or recommendation systems, where nodes have two different types (user and item), and links can exhibit multiple types, correlating with diverse ratings that users may assign to items. For knowledge graphs, link prediction is typically denoted as knowledge graph completion, where individual nodes represent different entities, and links represent multiple relations. Notably, it is often the case that a link prediction algorithm initially formulated for homogeneous graphs can be readily adapted to heterogeneous graphs, such as and knowledge graphs, through the incorporation of heterogeneous node type and relation type information.

### D.3 GRAPH NEURAL NETWORK

In recent years, there has been an increasing interest within the research community to apply graph structures with deep learning methods (Zhou et al., 2020; Hu et al., 2020). Graph Neural Networks (GNNs) have emerged as the most effective learning framework, demonstrating remarkable efficacy in addressing diverse tasks in a wide range of application domains.

Newly introduced neural network designs tailor for graph-structured data (Kipf & Welling, 2017; Velickovic et al., 2018; Schlichtkrull et al., 2018), have shown great performance across different domains. These areas encompass but are not limited to social networks and bioinformatics. It's worth highlighting that these innovative architectures have extended their influence into different research domains, including natural language processing (Zhang et al., 2019) and question answering (Huang et al., 2019).

### D.4 RELATIONAL CORRELATION GRAPH AND RELATIONAL CORRELATION NETWORK

TACT(Chen et al., 2021) first proposed Relational Correlation Graph(RCG) and Relational Correlation Network(RCN) to model semantic correlations between relations. It categorizes all pairs of relations into seven different topological patterns and turns original knowledge graphs into RCGs, which use relations to represent nodes and the seven different topological patterns to connect the nodes. Then, RCN is designed to capture the importance of different correlation patterns for inductive link prediction. The RCN module comprises two components: the correlation pattern component and the correlation coefficient component. The correlation pattern component considers the impact of different topological structures between relations, while the correlation coefficient component measures the degree of different correlations between relations.

Specifically, TACT aggregates the correlation coefficients of different correlation patterns for the relation $r_t$ to get the neighborhood embedding within a local extracted subgraph, which is denoted by $\mathbf{r}_t^N$.

$$\mathbf{r}_t^N = \frac{1}{6} \sum_{p=1}^{6} (\mathbf{N}_t^p \circ \mathbf{\Lambda}_t^p) \mathbf{R} \mathbf{W}^p \tag{2}$$

where $\mathbf{W}^p \in \mathbb{R}^{d \times d}$ is the weight parameter matrix, $\mathbf{R} \in \mathbb{R}^{|\mathcal{R}| \times d}$ denotes the embedding of all relations. Suppose the embedding of $r_i$ is $\mathbf{r}_i \in \mathbb{R}^{1 \times d}$, then $\mathbf{R}_{[i,:]} = \mathbf{r}_i$ where $\mathbf{R}_{[i,:]}$ denotes the $i^{th}$ slice along the first dimension. $\mathbf{N}_t^p \in \mathbb{R}^{1 \times |\mathcal{R}|}$ is the indicator vector where the entry $[\mathbf{N}_t^p]_i = 1$ if $r_i$ and $r_t$ are connected in the $p^{th}$ topological pattern, otherwise $[\mathbf{N}_t^p]_i = 0$. $\mathbf{\Lambda}_t^p \in \mathbb{R}^{1 \times |\mathcal{R}|}$ is the weight parameter, which indicates the degree of different correlations for the relation $r_t$ in the $p^{th}$ correlation pattern. Note that, we restrict $[\mathbf{\Lambda}_t^p]_i \geq 0$ and $\sum_{i=1}^{|\mathcal{R}|} [\mathbf{\Lambda}_t^p]_i = 1$.

Furthermore, TACT concatenates $\mathbf{r}_t$ and $\mathbf{r}_t^N$ to get the final embedding $\mathbf{r}_t^F$.

$$\mathbf{r}_t^F = \sigma([\mathbf{r}_t \oplus \mathbf{r}_t^N]\mathbf{H}) \tag{3}$$

where $\mathbf{H} \in \mathbb{R}^{2d \times d}$ is the weight parameters, and $\sigma$ is an activation function, such as $\mathrm{ReLU}(\cdot) = \max(0, \cdot)$. We call the module that models semantic correlations between relations as the *relational correlation module*, and $\mathbf{r}_t^F$ is the final output of the module.

### D.5 Relation Graph Convolutional Network

Relation Graph Convolutional Network(R-GCN)(Schlichtkrull et al., 2018) is used in many existing inductive link prediction methods. It use the following fomula to learn node embeddings:

$$\mathbf{e}_i^{(k+1)} = \sigma \left( \sum_{r \in \mathcal{R}} \sum_{j \in \mathcal{N}_i^r} \frac{1}{c_{i,r}} \mathbf{e}_j^{(k)} \mathbf{W}_r^{(k)} + \mathbf{e}_i^{(k)} \mathbf{W}_0^{(k)} \right)$$

where $\mathbf{e}_i^{(k)}$ denotes the embedding of entity $e_i$ of the $k^{th}$ layer in the R-GCN. $\mathcal{N}_i^r$ denotes the set of neighborhood indices of node $i$ under relation $r \in \mathcal{R}$. $c_{i,r} = |\mathcal{N}_i^r|$ is a normalization constant. $\mathbf{W}_r^{(k)} \in \mathbb{R}^{d \times d} (r \in \mathcal{R})$ and $\mathbf{W}_0^{(k)} \in \mathbb{R}^{d \times d}$ are the weight parameters. $\sigma(\cdot)$ is a activation function, such as the $\text{ReLU}(\cdot) = \max(0, \cdot)$.

## E More Details of Node Labeling

As mentioned in Section 2.3, after the extraction of the enclosing subgraph, the next step involves node labeling. For example, a technique referred to as Double Radius Node Labeling (DRNL) is used to assign integer labels to each node within the subgraph to give them with additional distance features. The node labeling process makes nodes more differentiable within the enclosing subgraph.

DRNL operates through the following steps. Initially, it assigns label $1$ to both node $x$ and node $y$. Then, any node $i$ characterized by a radius denoted as $(d(i, x), d(i, y)) = (1, 1)$ receives label $2$ and $(1, 2)$ or $(2, 1)$ are assigned with label $3$. Nodes with a radius of $(1, 3)$ or $(3, 1)$ are assigned label $4$. Nodes with a radius of $(2, 2)$ are assigned label $5$, and so forth.

In essence, DRNL incrementally assigns ascending labels to nodes based on their expanding radial distance from the two central nodes, $x$ and $y$.

## F More Discussion on CNMP

### F.1 Does the experimental setup seem difficult to apply in practice?

**A1**: We acknowledge the observation regarding the "1 vs. 50" setting adopted in CNMP's inductive link prediction, where 50 negative samples are randomly generated for each positive triple. It is important to note that this setting is not unique to CNMP but has its roots in prior works, such as GraIL (Teru et al., 2020), and is widely used in (Mai et al., 2021; Lin et al., 2022; Zhu et al., 2021; Xu et al., 2022).

Moreover, we would like to emphasize that the use of ranking strategy, i.e., the "1 vs. 50" setting, serves as a robust method for evaluating the model performance. This approach provides a representative subset of data that effectively captures the comprehensive ranking behavior of the model (Lebanon & Mao, 2007).

While we appreciate the concerns raised regarding the applicability of CNMP to the "1 vs. all" setting, we contend that the chosen experimental framework is well-suited for practical applications. Specifically, in scenarios such as product recommendation based on knowledge graphs and user recommendation relying on user graphs, inductive link prediction models, including CNMP, play an important role in accurately ranking a subset of nodes. It can be seen as a refining stage after ranking all entities to get more precise ranking results (Covington et al., 2016).

In conclusion, we maintain "1 vs. 50" experimental setting, which is both valid and practical. We believe that models assessed under this setting are practical for real-world applications and important for inductive link prediction.

### F.2 What are the advantages of $\text{CNMP}/\text{CNMP}_+$ computational graph compared to the simple method of extracting all paths between target nodes?

**A2**: The method of constructing a subgraph by simply extracting all paths between the target nodes such as, NBFNet (Zhu et al., 2021). These methods confront two major challenges. First, **scalability**.

The reasoning process of these methods needs to be conducted on the whole graph, making it difficult to generalize to large-scale KGs. Second, **efficiency**. For a path of length $n$ from node $u$ to node $v$, the computational complexity of these methods is $\mathcal{O}(\mathcal{D}^n)$, where $\mathcal{D}$ represents the average degree of nodes. However, for methods based on subgraphs, the computational complexity is $\mathcal{O}(2 \times \mathcal{D}^{\lfloor \frac{n+1}{2} \rfloor})$, where $\lfloor \rfloor$ denotes the ceiling function. When considering long-range relations, the subgraph-based methods can be more efficient. We also conduct comparisons in Table 5 to demonstrate the scalability of our CNMP method.

As demonstrated in Table 5, NBFNet confronts serious scalability issues in large KG. Specifically, NBFNet reasons on the whole KG, resulting in the out of memory issue. For both ILPC-small and ILPC-large datasets, GraIL and TACT both exhibit suboptimal performances. Conversely, CNMP consistently attains optimal results across both ILPC-small and ILPC-large datasets, which also demonstrates the effectiveness of our CNMP method.

### F.3 What is the difference between the current work and related works that are not discussed in related work section ?

**A3**: In order to elucidate our work more effectively, we discuss both the connection and the distinction between our work and related work:

1. **Connection with rule-based methods:** Our approach falls under GNN-based methods, while rule-based methods represent another classic branch of inductive link prediction tasks. Therefore, we include this category of methods in our related work section.

2. **Distinction from rule-based methods:** Rule-based methods are primarily concerned with the explicit learning of first-order logical rules, which limits their capacity to capture more complex semantic correlations between relations and affects their scalability to large knowledge graphs(Chen et al., 2021).

3. **Connection with GNN-based methods:** Our technical approach belongs to GNN-based methods. In our approach, we integrate two classic GNN-based methods (RCN and RGCN) with our proposed novel message passing strategy as a new scheme.

4. **Distinction from existing GNN-based methods:** Existing methods may lose key entities and relations during the extraction of enclosing subgraphs, leading to many disconnected reasoning paths. This severely hinders message passing in GNN-based methods, and consequently impacts overall reasoning performance. Our method, based on a novel message-passing mechanism, accurately extracts key information from enclosing subgraphs and outperforms existing SOTA methods in inductive link prediction.

### F.4 Does any statistical testing prove the effectiveness of the CNMP strategy ?

**A4**: We conduct experiments with statistical testing to demonstrate the effectiveness of the CNMP strategy. Specifically, we design experiments using three benchmarks, creating and varying levels of disconnection and noise in the graphs. Based on the statistics of the proportion of disconnected path and irrelevant information in this data, compare the experimental results.

As shown in Table 16 and Table 17 in the PDF of our global response, our method achieves notable performance enhancements even when confronted with graphs rampant with disconnected reasoning paths and irrelevant information. For instance, on the NELL-995 v1 dataset, where disconnection rates soar to 52.5%, CNMP achieved an improvement of nearly 4.82%. On the WN18RR v4 dataset, plagued by irrelevant information constituting 54.4%, CNMP led to a substantial increase of 17.7%. Furthermore, the results suggest that in most scenarios, the more severe the issues of disconnection and noise, the more pronounced the improvement potentially delivered by CNMP.

Table 16: The performance of triple classification tasks using the basic model RCN+RGCN under different disconnection rates. Note that EG represents the enclosing subgraph

| AUC-PR | WN18RR | | | | FB15k-237 | | | | NELL-995 | | | |
|---|---|---|---|---|---|---|---|---|---|---|---|---|
| | v1 | v2 | v3 | v4 | v1 | v2 | v3 | v4 | v1 | v2 | v3 | v4 |
| Disconnection Rates | 0.528 | 0.505 | 0.504 | 0.510 | 0.090 | 0.049 | 0.092 | 0.062 | 0.525 | 0.221 | 0.174 | 0.174 |
| 2-hop EG | 96.15 | 97.95 | 90.58 | 96.15 | 88.73 | 94.20 | 97.10 | 98.30 | 94.87 | 96.58 | 95.70 | 96.12 |
| 2-hop EG + CNMP | 99.27 (+3.12) | 98.41 (+0.46) | 93.90 (+3.32) | 99.27 (+3.12) | 93.97 (+5.24) | 97.40 (+3.20) | 98.83 (+1.73) | 99.39 (+1.09) | 99.69 (+4.82) | 99.17 (+2.59) | 99.30 (+3.60) | 99.07 (+2.95) |
| 2-hop EG + CNMP$_+$ | 99.14 (+2.99) | 99.77 (+1.82) | 97.75 (+7.17) | 99.94 (+3.79) | 92.54 (+3.81) | 96.45 (+2.25) | 99.66 (+2.56) | 99.43 (+1.13) | 99.95 (+5.08) | 99.92 (+3.34) | 99.98 (+4.28) | 99.56 (+3.44) |

Table 17: The performance of triple classification tasks using the basic model RCN+RGCN under different noise rates. Note that EG represents the enclosing subgraph

| AUC-PR | WN18RR | | | | FB15k-237 | | | | NELL-995 | | | |
|---|---|---|---|---|---|---|---|---|---|---|---|---|
| | v1 | v2 | v3 | v4 | v1 | v2 | v3 | v4 | v1 | v2 | v3 | v4 |
| Noise Rates | 0.580 | 0.501 | 0.575 | 0.544 | 0.615 | 0.787 | 0.798 | 0.830 | 0.989 | 0.918 | 0.883 | 0.917 |
| 3-hop EG | 97.79 | 96.43 | 88.15 | 81.57 | 88.34 | 94.42 | 97.16 | 98.16 | 93.95 | 95.97 | 93.83 | 94.76 |
| 3-hop EG + CNMP | 99.27 (+1.48) | 98.41 (+1.98) | 93.90 (+5.75) | 99.27 (+17.70) | 93.97 (+5.63) | 97.40 (+2.98) | 98.83 (+1.67) | 99.39 (+1.23) | 99.69 (+5.74) | 99.17 (+3.20) | 99.30 (+5.47) | 99.07 (+4.31) |
| 3-hop EG + CNMP$_+$ | 99.14 (+1.35) | 99.77 (+3.34) | 97.75 (+9.60) | 99.94 (+18.37) | 92.54 (+4.20) | 96.45 (+2.03) | 99.66 (+2.50) | 99.43 (+1.27) | 99.95 (+6.00) | 99.92 (+3.96) | 99.98 (+6.15) | 99.56 (+4.80) |

## F.5 WHAT IS THE DIFFERENCE BETWEEN OUR METHOD AND THE RULE-BASED EXISTING METHODS ?

**A5**: Our approach belongs to GNN-based methods rather than rule-based methods, so our approach does not learn specific rules. However, to help readers better understand our motivation, we introduce a rule-based learning perspective to explain it more clearly.

## F.6 HOW TO UNDERSTAND THE DIRECTIONALITY OF THE NEW RELATIONS IN THE RECONSTRUCTED COMPUTATIONAL GRAPH ?

**A6**: Regarding the establishment of the new equivalent relations section, how does CNMP define the directionality of these new relations? Like in Figure 3, why does $r_2'$ point to $u$ and not $v$ ?

With our method, $r_2'$ is an equivalent relation we have introduced, where the direction of the arrow is consistent with the original adjacent edge. Essentially, we borrow the direction of the original relation, which is why $r_2'$ points to $u$ and not $v$. However, please note that we do not strictly consider direction when modeling the reasoning path, and direction is only part of the relation information. This is because we do not wish to exclude some meaningful information due to directional constraints. For instance, among nodes $a_1$, $a_2$, and $a_3$, $a_2$ and $a_3$ are two sons of $a_1$ and both point to their father $a_1$. If we consider the directionality of relations, it would be impossible to extract an effective reasoning path between $a_1$ and $a_2$ to predict that their relation is one of brotherhood.

