# OpenReview forum: "Common Neighbor Induced Message Passing for Inductive Link Prediction in Knowledge Graphs"
_ICLR.cc/2025/Conference — ICLR 2025 Conference Withdrawn Submission_

### Official Review · Reviewer_Ls5q · 2024-10-30

**Soundness:** 1
**Presentation:** 2
**Contribution:** 1
**Rating:** 3
**Confidence:** 3

**Summary:**

This paper presents CNMP, a common neighbor induced message passing framework for inductive link prediction in KGs. Experiments across multiple datasets demonstrate that our method significantly outperforms existing state-of-the-art methods.

**Strengths:**

1. The paper is generally understandable and clearly explains the technical parts to a certain extent.
2. The figures and charts in the manuscript are exceptionally clear and well-presented.

**Weaknesses:**

1. This paper does not provide sufficient details on the THE CNMP+ STRATEGY.
2. All compared models listed in Table 3 and Table 4 are published in and before 2022, which are out of date. I strongly suggest the authors cite and compare with the methods published in 2023 and 2024.

**Questions:**

1. This paper does not provide sufficient details on the THE CNMP+ STRATEGY.
2. All compared models listed in Table 3 and Table 4 are published in and before 2022, which are out of date. I strongly suggest the authors cite and compare with the methods published in 2023 and 2024.

---

### Official Review · Reviewer_dKkn · 2024-11-02

**Soundness:** 2
**Presentation:** 2
**Contribution:** 2
**Rating:** 5
**Confidence:** 4

**Summary:**

This paper addresses issues of isolated entities and broken rule chains in current subgraph-based methods, proposing two solutions: CNMP, which mitigates the problem by embedding isolated nodes with shortest-path length information, and CNMP+, which fully restores broken paths. Experimental results show that both methods improve performance.

**Strengths:**

- This paper identifies and analyzes issues within existing subgraph-based methods.

- It proposes two solutions to mitigate and resolve these issues.

- Experimental results demonstrate the effectiveness of the proposed methods.

**Weaknesses:**

- This paper focuses on issues arising from the subgraph construction implementations of certain methods, rather than the methods as described in their corresponding papers. For instance, in GraIL’s code, both `np.max(dist)` and `np.sum(dist)` lead to issues. This paper primarily addresses these problems, though it does not clearly distinguish this context.

- The proof in Appendix A seems unnecessary, as it presents an obvious conclusion. Additionally, the proof lacks a condition specifying that if edge (A, r, B) exists, then edge (B, r_inverse, A) must also exist. Without this, since KGs are directed graphs, two nodes with a common neighbor may not have a connected path.

- Prior works, such as NBFNet, adopt the evaluation setting of treating all entities as candidates instead of the 50-negative setting. Reporting results with all entities as candidates would allow for a more reliable performance comparison.

- For the experiments in Section 3.5 and Table 6, fully-inductive datasets such as RMPI and "Ingram: Inductive Knowledge Graph Embedding via Relation Graphs, ICML 2023" are available. What was the motivation for constructing new datasets in this work?

- In the runtime experiment in Appendix B.8.3, only the smaller v1 dataset is used. A comparison on the larger v4 dataset would be more convincing. Additional details, such as whether the reported runtime includes preprocessing (especially for closed subgraph construction), are also needed.

- The source code and newly constructed dataset have not been provided, despite the authors' commitment to release them after the paper’s acceptance.

**Questions:**

Please see **Weaknesses** for details.

---

### Official Review · Reviewer_iJsz · 2024-11-04

**Soundness:** 3
**Presentation:** 2
**Contribution:** 2
**Rating:** 5
**Confidence:** 1

**Summary:**

The paper introduces the Common Neighbor Induced Message Passing (CNMP) framework for inductive link prediction in knowledge graphs, addressing the challenge of disconnected reasoning paths in subgraphs. CNMP enhances message passing by updating distance labels and introduces CNMP+ for iterative reasoning path reconstruction, demonstrating superior performance over state-of-the-art methods.

**Strengths:**

1.The proposed CNMP framework innovatively addresses the issue of disconnected reasoning paths in knowledge graphs, a problem that significantly hinders the effectiveness of message passing in GNN-based methods. The strategy to update distance labels for isolated common neighbors is a clever solution to ensure effective message passing.
2.This paper provides a thorough experimental evaluation across multiple datasets, showcasing the effectiveness of CNMP and CNMP+ strategies. The consistent outperformance against a range of baselines builds a strong case for the proposed methods.

**Weaknesses:**

1.Lack of comparision of the latest Inductive knowledge graph reasoning models, such C-MPNN[1], MINES[2], AdaProp[3] and InGram[4]. Most of baselines compared in this paper were published before 2022, which cannot show the effectiveness of the proposed model.
2.Although the author shows the algorithm running time comparison in the appendix, these experimental results are only conducted on some small-scale datasets. The author needs to provide a running time comparison on a large-scale dataset.
3.The problem of disconnected reasoning paths in the closed subgraph extraction process mentioned in the motivation of this article may be due to the discarding of some isolated nodes. If so, this will be very similar to the innovation of the paper [2], and the authors should discuss this more.

**Questions:**

Why only consider the cases with hop number 2 and 3? The author needs to discuss this in more detail in the motivation description.

---

### Official Review · Reviewer_JuVw · 2024-11-04

**Soundness:** 2
**Presentation:** 2
**Contribution:** 2
**Rating:** 3
**Confidence:** 4

**Summary:**

The paper presents an innovative solution to inductive KG reasoning. Specifically, the authors propose CNMP, a framework that leverages common neighbors to enhance message passing even with disconnected reasoning paths, and CNMP+, an enhanced version that improves entity and relation preservation. Extensive experiments have been conducted to verify the performance of the proposed model.

**Strengths:**

1. The introduction and related work sections are well organized. It clearly sketches the existing problem, the proposed argument, and related work for inductive KG completion.

2. Intensive experiments have been conducted to verify the performance of the proposed model.

**Weaknesses:**

1. The paper's motivation requires stronger justification. While the authors emphasize reasoning paths and propose methods to learn inductive patterns from common neighbors within the reasoning paths, they overlook crucial information contained in non-common neighbors. Knowledge in KGs often exists in diverse structural forms beyond simple reasoning paths - including tree-like and ego-graph-like structures. Consider an entity connected to multiple isolated neighbors through relations like 'born_in' and 'live_in'; these connections, though not part of a reasoning path, provide vital contextual information suggesting the entity is likely a person rather than a product. Furthermore, GNNs inherently excel at capturing such diverse knowledge structures through message passing. By focusing exclusively on common neighbors and reasoning paths, the proposed approach may unnecessarily constrain the model's ability to leverage the full spectrum of available knowledge patterns in the graph.

2. As the most crucial component of the inductive model, how to perform KG reasoning with unseen entities has not been well-analyzed in this paper. How could the model initialize or learn the embeddings of unseen entities? How does the model extract entity-independent features from the surrounding subgraphs for inductive KG reasoning?

3. The overall structure of this paper could be improved. The authors include several Q&As at the end of the Appendix. However, these do not further clarify the proposed model, as it is difficult to link them to the relevant sections of the main content. If the information in these Q&As is important, the authors should integrate them into the main body of the paper to provide better context and understanding.

4. More experiments are needed to verify the effectiveness of the proposed model. For instance, is there any case study to show how the model performs inductive reasoning based on the common neighbors?

5. Some experiment results are suspicious. For example, Tables 16 and 17 share the same result, even changing the 2-hop EG into 3-hop EG.

**Questions:**

See above.

---

### Note · Authors · 2024-11-25

I have read and agree with the venue's withdrawal policy on behalf of myself and my co-authors.